# Consumer preferences for telehealth in Australia: A discrete choice experiment

**Feby Savira**[1,2]*, **Suzanne Robinson**[1,3], **Kaylie Toll**[3], **Lauren Spark**[3], **Elizabeth Thomas**[3], **Julia Nesbitt**[4], **Isobel Frean**[5], **Richard Norman**[3]

**1** Deakin Health Economics, Institute for Health Transformation, School of Health and Social Development, Deakin University, Burwood, Victoria, Australia, **2** Global Centre for Preventive Health and Nutrition, Institute for Health Transformation, School of Health and Social Development, Deakin University, Burwood, Victoria, Australia, **3** School of Population Health, Faculty of Health Sciences, Curtin University, Perth, Western Australia, Australia, **4** Consumer Health Forum, Canberra, Australian Capital Territory, Australia, **5** Digital Health Cooperative Research Centre Limited (DHCRC), Sydney, New South Wales, Australia

* feby.savira@deakin.edu.au

**Data Availability Statement:** The dataset cannot be shared publicly because of ethics and data governance arrangement imposed by Curtin University Human Research Ethics Committee.

## Abstract

This study aims to elicit consumer preferences regarding telehealth and face-to-face consultations in Australia. It used a discrete choice experiment, presenting participants with a series of hypothetical choices, and based on their responses, infer what is most important to them. Data were analysed using conditional logit regression and latent class analysis. A total of 1,025 participants completed the survey, considering four different clinical scenarios. Face-to-face contacts were, on average, preferred to either telephone or video services. However, telehealth was identified as an attractive option if it prevents significant travel and can be conducted with a familiar doctor. Participants were strongly driven by cost, particularly greater than $30. Telehealth was least preferred for situations involving a new and unknown physical symptom, and relatively more preferred for surgical follow-up. The latent class analysis demonstrates only 15.9% of participants appeared unwilling to consider telehealth. The findings of this study suggest that meeting the needs of the Australian population requires a blended approach to service delivery, with telehealth being valued in a range of clinical scenarios. Price sensitivity was evident, therefore if telehealth services can be delivered with lower patient cost, then they are likely to be attractive.

## Introduction

Telehealth is a model of health service which involves remote interaction between service provider and patient, supported by information technologies [1]. This definition is broad and may include remote care services conducted with or without video connection, but largely refers to patient-provider interaction which occurs synchronously (i.e., in real-time). Telehealth has been widely advocated to supplement the face-to-face model of care particularly for people without easy access to health care services, such as those living in rural or remote areas, people with disability and/or reliant on caregivers. During COVID-19, telehealth use has increased significantly [2] and has played a critical role in enabling safe delivery of care during a global health emergency.

Please contact ROC-ethics@curtin.edu.au or (08) 9266 9223 to request access to the data.

**Funding:** This research was part of a larger study supported by the Australian Government Department of Health and Aged Care, Health Economics and Research Division, and the Digital Health Cooperative Research Centres Limited (DHCRC) (project DHCRC-0161). The DHCRC is funded under the Commonwealth's Cooperative Research Centres Program. The Department of Health and Aged Care had no role in study design, data collection and analysis, or preparation of the manuscript.

**Competing interests:** The other authors have declared that no competing interests exist.

In Australia, telehealth is not new. It was first rolled out in 2012 and mainly focused on specialist services and some select nursing services, with general practitioner (GP) telehealth services only introduced for the first time in November 2019 [3]. During this period, there were also geographic restrictions and user eligibility barriers in place [4], as well as poor digital literacy and awareness among patients and providers [5]. These resulted in a very low uptake among patients and providers–for example, in total, there were less than 100 GP telehealth consultations Australia-wide pre-COVID after the introduction of these items in November 2019 [6]. In response to the COVID-19 pandemic, the Australian Government expanded the subsidisation of telehealth Medicare items for all Australians. This policy aimed to ensure continuity of care by allowing care to be delivered remotely and to reduce COVID-19 transmission in the community [7]. The expansion of this subsidisation led to an immediate and substantial increase in use of telehealth across Australian primary care in the earlier phase of the pandemic [6], which has since largely stabilised [8].

Although descriptive and extensive across settings [9, 10], telehealth research to date particularly within the primary care setting lacks a strong evidence base, whereby benefits and concerns of consumers and clinicians have largely been assumed rather than based on strong evidence or experience. The lack of input from consumers in the current literature narrows the applicability of these findings and its translation for use in general telehealth practice [11]. Studies that have focused on consumer experience and preference have tended to do so in a unidimensional way that does not capture the complexity of consumer preferences [12–16]. There is a need to rigorously explore consumer preferences and attitudes towards telehealth in the primary care setting, to inform policy and ensure consumer engagement and sustainability of the health system [17]. Therefore, this study aims to elicit consumer preferences in relation to telehealth or face-to-face consultations using a discrete choice experiment.

## Methods

### Discrete choice experiment

Consumer preferences were determined using a discrete choice experiment (DCE). DCEs were initially widely used in marketing research, but have expanded considerably over recent years into academic research, including in Health Economics [18]. A DCE involves presenting participants with a series of hypothetical options, and based on their response, infer what aspects are most important to them. It adopts a Lancastrian approach in characterising how people make choices, in which choices are driven by a set of *dimensions* at particular *levels* [19, 20]. For example, an important dimension for telehealth might be cost, and the levels could be $0, $20, and $50.

The DCE presented as a series of *choice sets*, in this case, a series of forced pairwise choices posed to participants acting as a health service consumer. Participants face hypothetical scenarios wherein they were asked to imagine they are seeking healthcare, using three possible frames: needing a repeat prescription, having a new physical symptom, and requiring surgical follow-up. These frames were selected based on prior qualitative work, which have been described elsewhere [21]. The qualitative work suggested that people considered telehealth differently based on the reasons they were seeking care, hence we wanted to test this empirically in this way. An example choice task is presented in Fig 1.

Ethics approval was obtained from Curtin University Human Research Ethics Committee (HRE2021-0232).

**Imagine you need a repeat prescription for a condition you have.** To do this, you need to see the GP. There are two options available to you, which are described below. If you had to choose between these options, which would you pick?

| | Option A | Option B |
|---|---|---|
| How do you talk with the GP? | Telephone call | Video call |
| How far do you have to travel? | N/A | N/A |
| How well do you know the GP? | You have seen them once before | Very familiar |
| Does the GP have full access to your complete medical history? | Limited | No |
| How long do you have to wait before your appointment? | 5 days | 2 days |
| Technical quality (if relevant) | Small risk of distortion (5%) | Moderate risk of distortion (10%) |
| Opportunity to ask additional questions | Yes | No |
| Out of Pocket cost to you | $50 | Free |
| Which would you choose? | ○ Option A | ○ Option B |

next

**Fig 1. Example of a choice set.** In this example, the participant is being asked to consider a situation where they need a repeat prescription for a condition they have. To comply with the rules about plausible combinations of levels, both options do not involve travel, and do have a risk of distortion.

## Identification of dimensions and levels

Dimensions and levels were selected through an iterative process [22], combining qualitative data and clinical input from the research team. In brief, the qualitative work involved semi-structured focus group discussion regarding consumer attitudes and experiences with telehealth [21]. These focus groups were conducted by 10 trained facilitators with strong community networks. Participants were recruited by the trained facilitator through non-probability convenience sampling with each facilitator inviting up to 10 community members within their local community. This allowed health consumers, carers and community members who do not ordinarily participate in healthcare consultation to have their voice heard in a supportive and safe environment. A total of 90 participants from all across Australia contributed to the focus group discussions. Majority of the participants were female (79%) and aged between 35 to 54 (42%) or 55 to 74 (32%). Most participants were currently living in major cities, although a significant proportion lived in inner and outer regional areas of Australia, or had experience living in rural, regional or remote areas. Approximately half were living with a chronic health condition (47%). Seventy per cent of participants have used telehealth in the past. The analysis of the interview data employed a combination of inductive (generating new knowledge) and deductive (testing theories) techniques. Data were analysed sequentially according to the steps of thematic analysis detailed by Braun and Clarke [23]. The focus group discussions yielded a range of key themes regarding consumer attitudes towards telehealth, including convenience,

**Table 1. Selected dimensions and levels.**

| Dimension | KTD Theme* | Level 0 | Level 1 | Level 2 | Level 3 |
|---|---|---|---|---|---|
| How do you talk with the GP? | - | In-person | Telephone call | Video call | |
| How far do you have to travel? (if relevant) | Convenience and access | 5km from home | 20km from home | N/A | |
| How well do you know the GP? | Existing consumer-clinician relationship | Never seen before | You have seen them once before | Very familiar | |
| Does the GP have full access to your complete medical history? | Existing consumer-clinician relationship | No | Limited | Thorough | |
| How long do you have to wait before your appointment? | Wait time | 1 day | 3 days | 5 days | |
| Technical quality (if relevant) | Connectivity | N/A | Small risk of distortion (5%) | Moderate risk of distortion (10%) | |
| Opportunity to ask additional questions | Communication | No | Yes | | |
| Out-of-pocket cost to you | Cost | Free | $10 | $30 | $50 |

KTD: Kitchen Table Discussion.

*Details of Kitchen Table Discussion has been reported elsewhere [21].

access, fit-for-purpose, communication, existing consumer-clinician relationship, wait time, connectivity, and cost. A subset of these themes was identified through an expert panel discussion and implemented into the choice tasks (see Table 1).

## Design of choice tasks

The experimental design, which identifies which combinations of levels are seen together in each choice pair, was built using Ngene, a software designed for this specific purpose. The experimental design was underpinned by a large body of literature [24]. As we were interested in willingness to pay for services (and hence were looking at the trade-off between characteristics of the service and cost), we used C-efficiency as the prespecified design criteria to maximise efficiency of data collection and ensure unbiased and precise estimates of the ratios of coefficients [25]. The design consisted of 150 choice tasks divided into 10 blocks, each block containing 15 choice tasks that participants must complete. Small non-zero priors were implemented to account for prior beliefs around the direction of preferences across levels within dimensions, particularly those that have a natural order (for example, it is reasonable to assume that knowing the treating doctor is better than not knowing the doctor). Combinations of levels were checked for plausibility. This included not combining technical quality issues in a face-to-face consultation, or travel distance for a telephone of video consultation. To do this, we treated technical quality, mode of administration, and travel as a single dimension with each possible combination of those levels. Therefore, we considered six different combinations of these dimensions: face-to-face with 5 km travel; face-to-face with 20 km travel; telephone with small risk of distortion; telephone with moderate risk of distortion; video with small risk of distortion; and video with moderate risk of distortion. We also identified three frames (motivations) for seeking healthcare: needing a repeat prescription, having a new physical symptom, and requiring surgical follow-up. The frames were embedded into the 15 choice sets, with each participant completing 5 tasks related to each of the three frames.

## Survey implementation and participant consent

The design was implemented through Survey Engine, an international organisation which specializes in delivering DCE surveys [26]. The online survey was delivered to a panel of

participants who had indicated their interest in completing such surveys for small remuneration (approximately A$10). A sampling frame was imposed to ensure age and gender representativeness. In the survey, participants received information regarding objectives of the study, the structure of the survey, and were asked for their consent to participate. Participants who consented had their responses electronically recorded and were directed to complete the survey. Participants who did not provide consent were directed to exit the survey.

For participants who consented, they first answered screening questions regarding age and gender to ensure representativeness. Participants were then provided with the walkthrough and asked to complete 15 DCEs. Finally, they were asked to complete demographic questions (country of birth, postcode, primary language, education, general health question, presence of chronic condition) and two questions around experience with either telephone or video consultations, after which the survey ended. For participants who did not provide consent, they were immediately directed to exit the survey.

## Data analysis

The DCE data were analysed using conditional logit models and latent class analysis. Data analysis was conducted using Stata version 16 [27].

The conditional logit model is widely used for analysing state preference data [18, 28]. Conditional logit characterizes the mean preferences of the entire sample. In this study, a series of conditional logit models were performed, firstly using aggregate data from all responses obtained from all participants. Next, we conducted frame-specific analysis to identify variations in preferences around telehealth depending on reasons for seeking healthcare. Lastly, we analysed the data according to geographic location of the participant (metropolitan versus non-metropolitan areas). Postcode for each participant was matched to the corresponding Modified Monash Model (MMM) classification [29]; this is an Australia-specific measure of rurality. Analysis was conducted for two geographic classifications: MM1 (metropolitan) and MM2-7 (non-metro areas with increasing levels of rurality, wherein MM2 represents regional centres and MM7 represents very remote areas). This division was selected as the majority of Australian residents live in metropolitan regions and dividing the sample differently would lead to small sub-group sizes and therefore uncertainty around subsequent comparisons between groups.

To report findings in a way which is amenable to policy makers, we converted coefficients into willingness to pay estimates by taking their ratio relative to the coefficient on the linear term of cost in dollars. This approach allowed estimation of 95% confidence intervals [30, 31]. In the conditional logit context, the estimates represent the willingness to pay of the average participant to switch from the base (reference) level into another level.

The widely acknowledged drawback of the conditional logit analysis is that it assumes responses come from a common utility function (i.e., it does not reflect how preferences differ across the population). For example, if there is a small but significant minority of the population who are strongly resistant to using telehealth, that strength of preference would be diluted as the conditional logit only provides the mean preference. This has considerable policy implications especially in context of patient-centric care where multiple tailored models of care may be required. Therefore, to capture a range of community preferences, latent class analysis (LCA) was performed [32]. LCA is suitable to assess heterogeneity as it assumes there is a distinct number of classes (i.e., those with similar preferences) and predicts the probability of participants belonging to a particular class. The optimal number of classes was identified by repeating the regression using different numbers of classes, and then comparing the models based on face validity and model fit defined using Bayesian Information Criteria [33].

The most appropriate sample size for a DCE is largely undecided. Lancsar and Louviere [34] argued that 20 observations per choice set is sufficient to estimate reliable models. In this study, we have 102.5 observations per choice set (1025 respondents divided by 10 blocks of questions), suggesting that our results are likely to be robust. Several important themes emerged from the data and the large sample provides confidence in our results and allows for more detailed subgroup analysis.

Online data collection for this kind of study has significant benefits in terms of possible sample size, and geographical reach. However, it may be that the absence of an interviewer affects data quality. Therefore, a series of robustness checks was conducted on the conditional logit results. This was done through duplication of the conditional logit analysis but excluding (i) the fastest 10% survey completers; (ii) the slowest 10% completers; and (iii) participants who answered either A to every task, or B to every task.

## Results

### Participant characteristics

A total of 1025 participants completed the survey. The demographic characteristics of these respondents are reported in Table 2. The mean and median time to survey completion were respectively 9.4 and 7.7 minutes.

The sample distribution was representative of age and gender, as would be expected given this was imposed through the sampling frame. Most participants were Australia-born, with slightly higher proportion with at least Bachelor's education level or above, and predominantly reported very good or good health. Of participants, 56.2% and 17.3% had used telephone- and video-based consultations at least once in the last five years, respectively.

### Conditional logit models

Tables 3 & 4 details the results from the conditional logit in terms of coefficients and willingness to pay estimates. Negative coefficients indicate that the option is, on average, less preferred than the omitted base level. For the average survey participant, the characteristics of service delivery which most drove their choice were modality (face to face, video or telephone), and cost.

In the aggregate results (Table 3), each of the coefficient for participants preferred not to have to travel 20 km, and they would rather see their doctor face-to-face than via any telehealth options. Both video options were relatively less favoured than telephone options. Notably, the coefficient for face-to-face, 20 km travel (-0.336) is similar to telehealth options (ranging from -0.500 to -0.630). This suggests that if the travel requirement to see a doctor is more than 20 km, then the mean preference for telehealth would improve.

Regarding other dimensions, participants strongly preferred familiarity with the GP, and were cost sensitive. Time to wait before appointment also mattered but was of smaller magnitude.

Assessment of consumer preferences by the framing (repeat prescription, new symptom, and surgical follow-up) showed that the patterns between frames were largely similar to the aggregate findings. However, in terms of willingness-to-pay (reported in Fig 2 & Table 3), participants were less likely to choose any of the telehealth modalities if they need to see a doctor for assessment of a new physical symptom.

The conditional logit analysis based on geographic location is reported in Table 4. The non-metropolitan group appear more willing to travel further for a face-to-face consultation. Furthermore, they seem relatively less willing to engage in telehealth, either by telephone or video, but feel less strongly about waiting time than the metropolitan sample.

**Table 2. Demographic characteristics.**

| Characteristic | Level | Sample, n (%) |
|---|---|---|
| Gender | Male | 500 (48.8%) |
| | Female | 525 (51.2%) |
| | Gender diverse | 0 (0.0%) |
| | Would rather not say | 0 (0.0%) |
| Age (years) | 18–29 | 211 (20.6%) |
| | 30–39 | 183 (17.9%) |
| | 40–49 | 176 (17.2%) |
| | 50–59 | 169 (16.5%) |
| | 60–69 | 147 (14.3%) |
| | 70 and older | 139 (13.6%) |
| Country of birth | Australia | 802 (78.2%) |
| | Other English-speaking country | 114 (11.1%) |
| | Other non-English speaking country | 109 (10.6%) |
| Primary language | English | 963 (94.0%) |
| | Other | 62 (6.1%) |
| Highest level of education | Years 11 or below | 125 (12.2%) |
| | Year 12 | 150 (14.6%) |
| | Trade certificate | 139 (13.6%) |
| | Diploma | 150 (14.6%) |
| | Bachelor's degree | 333 (32.5%) |
| | Higher degree | 128 (12.5%) |
| General health rating | Excellent | 135 (13.2%) |
| | Very good | 366 (35.7%) |
| | Good | 330 (32.2%) |
| | Fair | 159 (15.5%) |
| | Poor | 35 (3.4%) |
| Chronic condition | Yes | 327 (31.9%) |
| | No | 698 (68.1%) |
| Use of telehealth in past five years | No | 449 (43.8%) |
| | 1–2 times | 362 (35.3%) |
| | 3 or more times | 214 (20.9%) |
| Use of videoconference in past five years | No | 848 (82.7%) |
| | 1–2 times | 137 (13.4%) |
| | 3 or more times | 40 (3.9%) |

## Latent class analysis

Latent class analysis was performed to account for heterogeneity. We explored latent class models using between 2 and 6 classes. Table 5 reports the results of the latent class analysis.

Based on both Bayesian Information Criterion and face validity of findings, the best solution was 4 classes. Class 1 (16.0% of the total sample) included participants with inconsistent preferences, that is, they differ from the average participant, and has some unexpected coefficients. Participants belonging in this class were also willing to pay higher out of pocket costs and prefers seeing a GP they do not know. Class 2 (15.9%) included those whose preferences were primarily driven by face-to-face options. They were somewhat sensitive to price, albeit much smaller than the preference for face-to-face options. Class 3 (34.8%) included participants who were willing to engage with telehealth, strongly preferred not having to travel more

**Table 3. Results from conditional logit models, overall and by frame.**

| Dimension | Level | Overall (n = 1,025) | | By frame | | | | | |
|---|---|---|---|---|---|---|---|---|---|
| | | | | Repeat prescription | | New physical symptom | | Surgical follow-up | |
| | | Coefficient (SE) | WTP ($, 95% CI) | Coefficient (SE) | WTP ($, 95% CI) | Coefficient (SE) | WTP ($, 95% CI) | Coefficient (SE) | WTP ($, 95% CI) |
| How do you talk with the GP? | In-person (5 km travel) | Reference level | | Reference level | | Reference level | | Reference level | |
| | In-person (20 km travel) | -0.336 (0.046) *** | -11 (-14, -8) | -0.466 (0.078) *** | -14 (-19, -9) | -0.367 (0.075) *** | -13 (-18, -8) | -0.195 (0.074) *** | -7 (-11, -2) |
| | Telephone call (5% distortion) | -0.500 (0.046) *** | -15 (-18, -12) | -0.377 (0.079) *** | -9 (-14, -4) | -0.686 (0.081) *** | -22 (-28, -17) | -0.448 (0.075) *** | -14 (-19, -9) |
| | Telephone call (10% distortion) | -0.584 (0.048) *** | -20 (-23, -17) | -0.532 (0.076) *** | -18 (-23, -13) | -0.693 (0.081) *** | -24 (-30, -19) | -0.550 (0.077) *** | -19 (-24, -14) |
| | Video call (5% distortion) | -0.551 (0.052) *** | -18 (-21, -14) | -0.615 (0.084) *** | -19 (-24, -13) | -0.672 (0.085) *** | -23 (-28, -17) | -0.410 (0.081) *** | -13 (-19, -8) |
| | Video call (10% distortion) | -0.630 (0.053) *** | -20 (-24, -16) | -0.669 (0.081) *** | -20 (-25, -14) | -0.736 (0.083) *** | -24 (-30, -19) | -0.525 (0.078) *** | -17 (-22, -12) |
| How well do you know the GP? | Never seen before | Reference level | | Reference level | | Reference level | | Reference level | |
| | Once before | 0.275 (0.029) *** | 10 (8, 12) | 0.247 (0.048) *** | 8 (5, 12) | 0.282 (0.050) *** | 10 (7, 13) | 0.292 (0.046) *** | 11 (8, 14) |
| | Very familiar | 0.557 (0.034) *** | 19 (17, 21) | 0.482 (0.051) *** | 16 (13, 19) | 0.589 (0.051) *** | 21 (17, 24) | 0.617 (0.052) *** | 20 (17, 23) |
| GP access to medical history | No | Reference level | | Reference level | | Reference level | | Reference level | |
| | Limited | 0.064 (0.028)** | 1 (0, 3) | -0.018 (0.046) | -1 (-4, 2) | 0.139 (0.046) *** | 4 (1, 7) | 0.074 (0.047) | 1 (-2, 4) |
| | Thorough | 0.237 (0.027) *** | 7 (5, 9) | 0.305 (0.046) *** | 9 (6, 12) | 0.233 (0.047) *** | 8 (4, 11) | 0.181 (0.046) *** | 5 (2, 8) |
| Time to wait before appointment | 1 day | Reference level | | Reference level | | Reference level | | Reference level | |
| | 3 days | -0.082 (0.027) *** | -2 (-4, 0) | -0.142 (0.049) *** | -3 (-6, 0) | -0.079 (0.047) | -2 (-5, 1) | -0.029 (0.047) | -1 (-4, 2) |
| | 5 days | -0.314 (0.029) *** | -10 (-12, -8) | -0.410 (0.050) *** | -12 (-15, -9) | -0.319 (0.049) *** | -10 (-14, -7) | 0.229 (0.049) *** | -7 (-10, -4) |
| Opportunity to ask questions | No | Reference level | | Reference level | | Reference level | | Reference level | |
| | Yes | 0.124 (0.023) *** | 5 (4, 7) | 0.047 (0.036) | 3 (1, 5) | 0.201 (0.035) *** | 8 (5, 10) | 0.126 (0.037) *** | 5 (3, 7) |
| Out-of-pocket cost to you | $0 | Reference level | | Reference level | | Reference level | | Reference level | |
| | $10 | -0.063 (0.062) | - | -0.242 (0.113) ** | - | -0.121 (0.109) | - | -0.147 (0.119) | - |
| | $30 | -0.994 (0.067) *** | - | -1.234 (0.115) *** | - | -0.973 (0.111) *** | - | -0.817 (0.118) *** | - |
| | $50 | -1.310 (0.047) *** | - | -1.301 (0.063) *** | - | -1.319 (0.060) *** | - | -1.335 (0.063) *** | - |

WTP: willingness to pay.

Statistical significance is indicated at the 1% (***), 5% (**) and 10% (*) level. Willingness-to-pay analysis was performed by enforcing constant utility of dollars.

than 5 km and were very price sensitive (i.e., will almost always select option with lower cost). Class 4 (33.3%) included participants with mixed preference, similar to the overall survey participants. These participants were willing to engage with telehealth but prefers face-to-face option. They also favour seeing familiar doctor, with some cost sensitivity but less than Class 3.

## Clarity and robustnesss

In terms of clarity, 89.7% of respondents indicated the task was clear or very clear, 3.1% found the task unclear, and the remainder were neutral. Regarding difficulty, 78.3% indicated the task was easy or very easy, 5.5% found it difficult and the remainder found it neither easy nor difficult. This suggests majority of respondents were able to engage with the task and

**Table 4. Results from conditional logit models, by location.**

| Dimension | Level | By location | | | |
|---|---|---|---|---|---|
| | | Metropolitan (n = 795) | | Non-metropolitan (n = 230) | |
| | | *Coefficient (SE)* | *WTP ($, 95% CI)* | *Coefficient (SE)* | *WTP ($, 95% CI)* |
| How do you talk with the GP? | In-person (5 km travel) | Reference level | | Reference level | |
| | In-person (20 km travel) | -0.356 (0.052)*** | -12 (-16, -9) | -0.258 (0.098)*** | -8 (-13, -3) |
| | Telephone call (5% distortion) | -0.463 (0.051)*** | -14 (-18, -11) | -0.645 (0.109)*** | -17 (-23, -10) |
| | Telephone call (10% distortion) | -0.541 (0.054)*** | -19 (-23, -16) | -0.785 (0.106)*** | -22 (-28, -16) |
| | Video call (5% distortion) | -0.529 (0.056)*** | -18 (-22, -14) | -0.641 (0.127)*** | -18 (-25, -11) |
| | Video call (10% distortion) | -0.588 (0.059)*** | -19 (-24, -15) | -0.83 (0.124)*** | -22 (-29, -15) |
| How well do you know the GP? | Never seen before | Reference level | | Reference level | |
| | Once before | 0.272 (0.032)*** | 10 (8, 12) | 0.297 (0.067)*** | 9 (6, 12) |
| | Very familiar | 0.514 (0.038)*** | 18 (16, 21) | 0.735 (0.076)*** | 20 (16, 25) |
| GP access to medical history | No | Reference level | | Reference level | |
| | Limited | 0.066 (0.030)** | 2 (0, 4) | 0.049 (0.067) | 0 (-3, 4) |
| | Thorough | 0.214 (0.030)*** | 7 (5, 9) | 0.32 (0.063)*** | 8 (4, 12) |
| Time to wait before appointment | 1 day | Reference level | | Reference level | |
| | 3 days | -0.087 (0.031)*** | -2 (-5, 0) | -0.051 (0.06) | -1 (-4, 3) |
| | 5 days | -0.335 (0.033)*** | -11 (-14, -9) | -0.231 (0.061)*** | -6 (-9, -2) |
| Opportunity to ask questions | No | Reference level | | Reference level | |
| | Yes | 0.124 (0.026)*** | 5 (4, 7) | 0.121 (0.048)** | 5 (2, 7) |
| Out-of-pocket cost to you | $0 | Reference level | | Reference level | |
| | $10 | -0.100 (0.070) | - | 0.099 (0.136) | - |
| | $30 | -0.968 (0.074)*** | - | -1.084 (0.155)*** | - |
| | $50 | -1.258 (0.052)*** | - | -1.525 (0.11)*** | - |

WTP: willingness to pay.

Statistical significance is indicated at the 1% (***), 5% (**) and 10% (*) level. Willingness-to-pay analysis was performed by enforcing constant utility of dollars.

understood what was required of them. The robustness test results showed there was no nota-ble difference compared to the aggregate conditional logit findings.

## Discussion

This study explored community preferences for telehealth services using a discrete choice experiment. Although based on an Australian sample, we believe the general conclusions are important for other countries at a similar stage of digital health transformation [35]. Our anal-ysis demonstrated the average Australians appeared to favour face-to-face consultations over telehealth options. This finding is echoed in a recent cross-sectional survey [36]. However, as the required distance to travel increased, the DCE revealed preference for face-to-face became much weaker, suggesting a major potential role for telehealth in circumstances where primary care services are located further from the community. Participants were price sensitive and much less willing to select hypothetical choices that required a co-payment of $30 or $50. This suggests that, if telehealth can be delivered to communities without any out-of-pocket fee, then consumers appear open to using such services.

While telehealth is not a 'one-size-fits-all' model of care, the DCE findings indicated appro-priateness of telehealth at certain points of the patient journey. For example, consumers noted that telehealth was appropriate for some routine consultations such as repeat prescriptions, referral or follow up appointments. Consumers further suggested that telehealth was not as

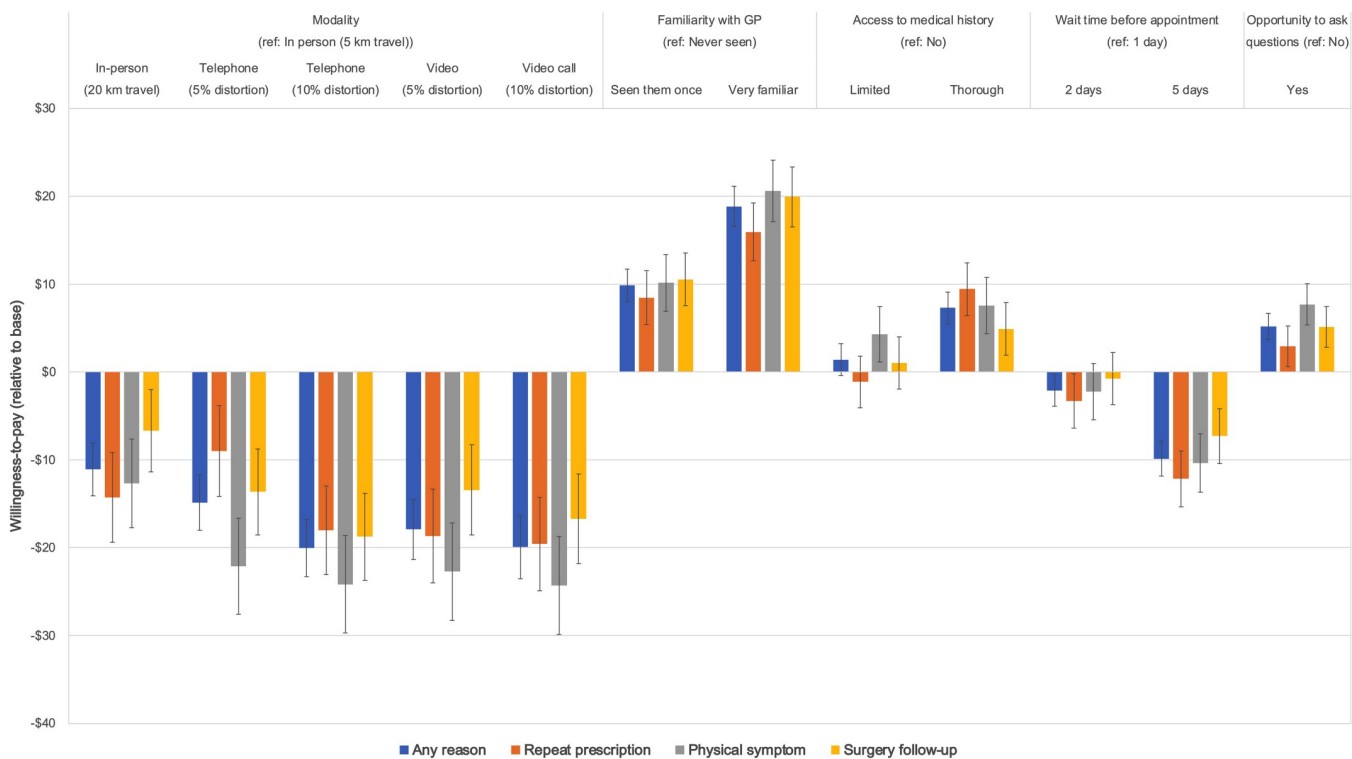

**Fig 2. Willingness to pay threshold analysis, by frame.**

appropriate for more complex or sensitive issues or where a physical examination is required, consistent with other studies [36]. Having existing relationships with a trusted and familiar clinician was also highly regarded. This relates to the continuity aspects of good primary care which is important across time, settings, conditions and people. Primary care should be the lynchpin for continuity of care across all stages of life [37], and telehealth incorporated into an effective model of primary care could support this.

The research did not identify a consistent strong preference between video and telephone contacts, although there is limited evidence in favour of telephone. Recent analysis of telehealth mental health services revealed that technical and connectivity issues contributed to telephone being the preferred modality [38]. The clinician aspect, which was not explored in this study, should also be considered. For example, a recent qualitative study revealed Australian GPs preferred telephone consultations because it is simpler to do, and the Medicare rebate was the same as video consultations [39].

The latent class analysis revealed that there is considerable difference in preferences across the population. There is a small group that strongly favour face-to-face consultations, but the majority of participants appeared to be accepting of other modalities, if other characteristics of their consultation (cost, familiarity with their doctor) were favourable. Regarding out-of-pocket expenses incurred by consumers, this is more likely the case for video consultation, which may put a strain on Wi-Fi and mobile data costs and those with 'pay as you go' internet and data arrangements. The inequitable access of telehealth is of high concern for those groups with limited or no access to modern communication technology [40].

Consumers living in rural and regional Australia preferred travelling out to see their doctor and were less likely to engage with any telehealth modalities. This contrasts a recent study reporting Australians living in rural and regional areas were less likely to prioritise in-person

**Table 5. Latent class analysis.**

| Dimension | Level | Class 1 (33.3%) | Class 2 (16.0%) | Class 3 (34.8%) | Class 4 (15.9%) |
|---|---|---|---|---|---|
| | | *Coefficient* | *Coefficient* | *Coefficient* | *Coefficient* |
| How do you talk with the GP? | In-person (5km travel) | Reference | Reference | Reference | Reference |
| | In-person (20km travel) | -0.636 | -0.092 | -0.777 | -0.268 |
| | Telephone call (5% distortion) | -0.534 | -0.074 | -0.649 | -2.005 |
| | Telephone call (10% distortion) | -0.597 | -0.049 | -0.831 | -2.230 |
| | Video call (5% distortion) | -0.289 | -0.055 | -0.687 | -2.884 |
| | Video call (10% distortion) | 0.454 | -0.067 | -0.769 | -3.267 |
| How well do you know the GP | Never seen before | Reference | Reference | Reference | Reference |
| | Once before | 0.686 | -0.059 | 0.259 | 0.265 |
| | Very familiar | 1.309 | 0.014 | 0.481 | 0.545 |
| GP access to medical history | No | Reference | Reference | Reference | Reference |
| | Limited | 0.257 | 0.062 | 0.094 | -0.042 |
| | Thorough | 0.586 | 0.066 | 0.180 | 0.198 |
| Time to wait before appointment | 1 day | Reference | Reference | Reference | Reference |
| | 3 days | -0.044 | -0.158 | -0.241 | -0.038 |
| | 5 days | -0.497 | -0.252 | -0.407 | -0.355 |
| Opportunity to ask questions | No | Reference | Reference | Reference | Reference |
| | Yes | 0.176 | 0.174 | 0.169 | 0.060 |
| Out-of-pocket cost to you | $0 | Reference | Reference | Reference | Reference |
| | $10 | -0.151 | -0.270 | -0.265 | 0.259 |
| | $30 | -0.843 | -0.150 | -2.943 | -0.509 |
| | $50 | -1.392 | 0.248 | -4.044 | -1.326 |

Class 1: mixed preference group, similar to average; Class 2: inconsistent preferences; Class 3: no travel, very price sensitive; Class 4: strong face-to-face focus.

consultation, however noting this was a survey conducted in tertiary hospitals [36]. Our findings may reflect a lack of familiarity as well as poorer digital health literacy education in rural and regional communities [5]. Indeed in a study of a highly educated and digitally literate population, consumers proposed 68% of in-person consultations could be replaced with video consultations [41]. This suggests that there is a need to first address the barriers to using health technologies and improve infrastructure in rural and remote communities. Rural and regional Australia are home to many vulnerable communities with often poorer access to primary care [42]. If the foundation to using telehealth are well established, telehealth could support increased access to care for our most vulnerable.

## Policy, practice, and research implications

Telehealth services should not be full substitutes for face-to-face care, and should ideally be linked to provision by a provider with a prior relationship with the patient. Exceptional circumstances where it is not possible for patients to have an existing relationship with a particular provider should be catered for to ensure equity of access. Targeting rural and remote populations would allow for more detailed exploration of geographical differences in research i.e., the different value of telehealth for rural/remote versus urban/metro consumers.

For many telehealth services, Medicare funding is temporary. As the pandemic abates, policy makers have several considerations if the community is to see telehealth services continue. Continuation needs to occur in a manner that avoids low value for money, and that provide high value technologies and experiences of care. The increased scope of availability and

appointment times will also allow for timely access to medical care. Addressing issues early and within primary care could increase the focus on 'right care, right time, right place'.

## Limitation

A limitation of the study is that it was conducted in an internet panel of respondents. The topic of the DCE was partly around the willingness of the community to engage with telehealth, and it might be reasonable to hypothesise that online participants would be relatively more engaged with online services. Furthermore, the DCE was only available to those who can access the internet. The sample was also slightly biased to participants with at least a first-degree education (33%). Educational attainment has been shown to be positively related to willingness to use technology [43]. Finally, the DCE survey was delivered to a panel of participants for a small remuneration, leading to the possibility of bias. Nonetheless, this paper provides important insights into the community preferences of telehealth in Australia, which has important policy implications. It would be useful to expand the research to include the voice of those less likely to opt into panels of survey respondents, such as those who are digitally limited, those with lower education levels, culturally and linguistically diverse and other disadvantaged population groups. Future studies should consider conducting the survey using different recruitment strategies, for example through face-to-face means in the community.

## Conclusion

In conclusion, the findings of this study suggest Australians are keen to have a blended approach to receiving services, with telehealth being highly valued by a significant proportion of the respondents, in a range of clinical scenarios. While face-to-face contacts were, on average, preferred to either telephone or video services, virtual care was identified as an attractive option if it prevents significant travel, and can be conducted with a doctor that the patient knows. Finally, cost was a significant driver of choice. If telehealth services can be delivered with minimum out of pocket expenses, then they become an attractive option.

## Acknowledgments

We thank participants for their time and commitment to completing the survey.

## Author Contributions

**Conceptualization:** Suzanne Robinson, Richard Norman.

**Data curation:** Lauren Spark.

**Formal analysis:** Feby Savira, Richard Norman.

**Funding acquisition:** Suzanne Robinson, Isobel Frean, Richard Norman.

**Investigation:** Kaylie Toll, Elizabeth Thomas, Julia Nesbitt.

**Methodology:** Feby Savira, Suzanne Robinson, Richard Norman.

**Project administration:** Kaylie Toll, Elizabeth Thomas, Julia Nesbitt, Richard Norman.

**Resources:** Suzanne Robinson, Julia Nesbitt, Isobel Frean.

**Software:** Richard Norman.

**Supervision:** Suzanne Robinson.

**Validation:** Suzanne Robinson, Kaylie Toll, Lauren Spark, Elizabeth Thomas, Richard Norman.

**Visualization:** Feby Savira.

**Writing – original draft:** Feby Savira, Suzanne Robinson, Richard Norman.

**Writing – review & editing:** Feby Savira, Suzanne Robinson, Kaylie Toll, Lauren Spark, Elizabeth Thomas, Julia Nesbitt, Isobel Frean, Richard Norman.

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
