## [Decision Letter · Decision Letter 0]

19 Dec 2022

PONE-D-22-29056Consumer preferences for telehealth in Australia: a discrete choice experimentPLOS ONE

Dear Dr. Savira,

Thank you for submitting your manuscript to PLOS ONE. After careful consideration, we feel that it has merit but does not fully meet PLOS ONE’s publication criteria as it currently stands. Therefore, we invite you to submit a revised version of the manuscript that addresses the points raised during the review process. The contribution merits to be considered for publication but as indicated by the reviewers it needs to address some aspects related to the implementation of the research design. Specifically, there is a need to clary aspects related to  data collection using semi-structured focus group, the process of theme identification for DCE survey, sampling strategy and possible biases, presentation of choice sets. 

We look forward to receiving your revised manuscript.

Kind regards,

Francesca Ferrè

Academic Editor

PLOS ONE

Journal Requirements:

“FS is supported by an Alfred Deakin Postdoctoral Research Fellowship from Deakin University.”

“This research was part of a larger study supported by the Australian Government Department of Health and Aged Care, Health Economics and Research Division, and the Digital Health Cooperative Research Centres Limited (DHCRC) (project DHCRC-0161). The DHCRC is funded under the Commonwealth’s Cooperative Research Centres Program. The Department of Health and Aged Care had no role in study design, data collection and analysis, or preparation of the manuscript.”

Additional Editor Comments:

The contribution provides an empirical analysis about population preferences around telehealth services in Australia.

The topic presented and the quantitative analysis and evidence are important to inform current debate about the implementation and use at large scale of telehealth solutions, thus I believe the contribution fits the purpose of the journal.

I have few concerns about the research design applied, specifically authors should:

I) provide additional information about the composition and numbers of semi-structured focus group(s) discussion done to identify main themes for the DCE;

II) clarify the process for the identification of sub-set of themes: what analytical approach has been followed (eg thematic analysis or similar?);

III) discuss in the limitations the possible distortion/bias of respondent to the DCE since it is said survey respondents were given a pay.

Reviewers' comments:

Reviewer's Responses to Questions

**Comments to the Author**

1. Is the manuscript technically sound, and do the data support the conclusions?

Reviewer #1: Yes

2. Has the statistical analysis been performed appropriately and rigorously? 

Reviewer #1: Yes

3. Have the authors made all data underlying the findings in their manuscript fully available?

Reviewer #1: No

4. Is the manuscript presented in an intelligible fashion and written in standard English?

Reviewer #1: Yes

5. Review Comments to the Author

Reviewer #1: The paper aims to elicit the patients’ preferences on telehealth and face-to-face consultations in Australia. The authors intend to understand which are the potential barriers and facilitators that could discourage or encourage patients to prefer remote visits than in-person visits, respectively. To address this question, the authors implement a Discrete Choice Experiment, where respondents identify their preferred option between two alternative medical consultations, which differ with respect to some characteristics (attributes).

I think this work represents an innovative contribution in the medical literature on telehealth. The research question is highly interesting, because, although during the COVID-19 pandemic the potential benefits of using telehealth (together with all the other remote services) clearly emerged, the actual usage of these services is still limited, and their effective diffusion has not been quantified yet. The DCE experiment is well planned, and results are highly interesting from the policy making perspective. In addition, I find it very useful to differentiate the analysis among the different reason that motivate people to seek healthcare services.

However, some aspects that could still be improved by the authors.

First, the process for identifying the attributes and levels to be used in the DCE is not extensively discussed, in my opinion. As far as I understand, the authors combined the qualitative information that results from a focus group discussion, and some other clinical input. I would personally like to know more details on the focus group discussion: how many participants were involved? how were the participants chosen? which were their characteristics (age, gender, professional experience)? which were the elements that emerged from the discussion? How did the authors identify the exact subset of attributes (and levels) to be employed in the DCE?

Second, the sampling strategy to deliver the web survey does not follow a probabilistic rule (the survey has been delivered to a panel of participants who declared to be willing to participate to the survey for small remuneration). I would more extensively discuss this limitation in the paper, while advancing the policy implications of the study.

Third, it is not clear to me how do you account for the different motivations that can push patients to seek healthcare services (needing a repeat prescription, having a new physical symptom, and requiring surgical follow-up). Does the experiment “assign” each respondent to one of these frames so that the 15 choice sets that the given respondent faces are all related to the frame he/she is actually assigned to? I strongly suggest clarifying this issue in the manuscript.

6. PLOS authors have the option to publish the peer review history of their article (what does this mean?). If published, this will include your full peer review and any attached files.

Reviewer #1: **Yes: **Costanza Tortù

---

## [Author Response · Author response to Decision Letter 0]

1 Mar 2023

Additional Editor Comments:

The contribution provides an empirical analysis about population preferences around telehealth services in Australia. The topic presented and the quantitative analysis and evidence are important to inform current debate about the implementation and use at large scale of telehealth solutions, thus I believe the contribution fits the purpose of the journal.

I have few concerns about the research design applied, specifically authors should:

I) provide additional information about the composition and numbers of semi-structured focus group(s) discussion done to identify main themes for the DCE;

• Response: The focus group discussions were conducted by 10 community members, each hosting a group of up to 10 participants. A total of 90 participants took part from across Australia, and 70% reported using telehealth in the past. We have added this information in the Methods section (page 5-6). Please also see our response to Reviewer to the first set of questions below as it is related to the Editor’s question.

II) clarify the process for the identification of sub-set of themes: what analytical approach has been followed (eg thematic analysis or similar?);

• Response: The sub-set of themes were identified through expert panel discussion with the team. The identified dimensions and constituent levels we identified as being of broad importance across the focus group discussion. However, the interview data was analysed using qualitative methods. See below for the updated Methods section (page 5-6) and refer to our previous publication for more details (PLoS One. 2022; 17(8): e0273935).

“The analysis of the interview data employed a combination of inductive (generating new knowledge) and deductive (testing theories) techniques. Data were analysed sequentially according to the steps of thematic analysis detailed by Braun and Clarke. The focus group discussions yielded a range of key themes regarding consumer attitudes towards telehealth, including convenience, access, fit-for-purpose, communication, existing consumer-clinician relationship, wait time, connectivity, and cost. We identified a subset of these themes to implement into the choice tasks based on relative importance.”

III) discuss in the limitations the possible distortion/bias of respondent to the DCE since it is said survey respondents were given a pay.

• Response: We thank the Editor for the feedback. Please refer to our response to Reviewer’s second question below. 

 

Reviewers’ comments:

Reviewer #1: The paper aims to elicit the patients’ preferences on telehealth and face-to-face consultations in Australia. The authors intend to understand which are the potential barriers and facilitators that could discourage or encourage patients to prefer remote visits than in-person visits, respectively. To address this question, the authors implement a Discrete Choice Experiment, where respondents identify their preferred option between two alternative medical consultations, which differ with respect to some characteristics (attributes).

I think this work represents an innovative contribution in the medical literature on telehealth. The research question is highly interesting, because, although during the COVID-19 pandemic the potential benefits of using telehealth (together with all the other remote services) clearly emerged, the actual usage of these services is still limited, and their effective diffusion has not been quantified yet. The DCE experiment is well planned, and results are highly interesting from the policy making perspective. In addition, I find it very useful to differentiate the analysis among the different reason that motivate people to seek healthcare services.

However, some aspects that could still be improved by the authors.

First, the process for identifying the attributes and levels to be used in the DCE is not extensively discussed, in my opinion. As far as I understand, the authors combined the qualitative information that results from a focus group discussion, and some other clinical input. I would personally like to know more details on the focus group discussion: how many participants were involved? How were the participants chosen? Which were their characteristics (age, gender, professional experience)? Which were the elements that emerged from the discussion? How did the authors identify the exact subset of attributes (and levels) to be employed in the DCE?

• Response: We have now added more information to the Methods section to clarify these details. 

“In brief, the qualitative work involved semi-structured focus group discussion regarding consumer attitudes and experiences with telehealth [21]. These focus groups were conducted by 10 trained facilitators with strong community networks. Participants were recruited by the trained facilitator through non-probability convenience sampling with each facilitator inviting up to 10 community members within their local community. This allowed health consumers, carers and community members who do not ordinarily participate in healthcare consultation to have their voice heard in a supportive and safe environment. A total of 90 participants from all across Australia contributed to the focus group discussions. Majority of the participants were female (79%) and aged between 35 to 54 (42%) or 55 to 74 (32%). Most participants were currently living in major cities, although a significant proportion lived in inner and outer regional areas of Australia, or had experience living in rural, regional or remote areas. Approximately half were living with a chronic health condition (47%). Seventy per cent of participants have used telehealth in the past.” (Methods, page 5-6).

Regarding subset of attributes, this has been answered in response to the Editor’s question (II). 

Second, the sampling strategy to deliver the web survey does not follow a probabilistic rule (the survey has been delivered to a panel of participants who declared to be willing to participate to the survey for small remuneration). I would more extensively discuss this limitation in the paper, while advancing the policy implications of the study.

• Response: We have added this point to the Limitations section. Note that we had age- and gender- sampling frames imposed (now added in Methods, page 8, and already discussed under Table 2, page 12-13), thus we believe the sampling strategy is probabilistic in those dimensions.

“Finally, the DCE survey was delivered to a panel of participants for a small remuneration, leading to the possibility of bias. Nonetheless, this paper provides important insights into the community preferences of telehealth in Australia, which has important policy implications. It would be important to expand the research to include the voice of those less likely to opt into panels of survey respondents, such as those who are digitally limited, those with lower education levels, culturally and linguistically diverse and other disadvantaged population groups. Future studies should also consider conducting the survey using different recruitment strategies, for example through face-to-face means in the community. (Limitations section, page 23).

Third, it is not clear to me how do you account for the different motivations that can push patients to seek healthcare services (needing a repeat prescription, having a new physical symptom, and requiring surgical follow-up). Does the experiment “assign” each respondent to one of these frames so that the 15 choice sets that the given respondent faces are all related to the frame he/she is actually assigned to? I strongly suggest clarifying this issue in the manuscript.

• Response: This is a very important point, and the prior qualitative point suggested this to be the case here. The three frames (motivations) for seeking healthcare ie., needing a repeat prescription, having a new physical symptom, and requiring surgical follow-up, were all embedded into the 15 choice sets, with each participant completing 5 tasks related to each of the three frames. This information has now been added to the Methods section, page 8. 

Interestingly, the frames did not demonstrate a very strong pattern of difference between frames, which was one of the few areas where the qualitative and quantitative results did not align. We do not have a hypothesis for such a difference which can be tested using current data, but it is certainly worthy of follow-up in future work.

---

## [Decision Letter · Decision Letter 1]

20 Mar 2023

Consumer preferences for telehealth in Australia: a discrete choice experiment

PONE-D-22-29056R1

Dear Dr. Savira,

We’re pleased to inform you that your manuscript has been judged scientifically suitable for publication and will be formally accepted for publication once it meets all outstanding technical requirements.

Kind regards,

Francesca Ferrè

Academic Editor

PLOS ONE

Additional Editor Comments (optional):

We thanks the authors for reviewing the manuscript. With their revision, the authors have taken into account all the comments, which has led to an improved manuscript that can now be coinsedered for publication.

Reviewers' comments:

Reviewer's Responses to Questions

**Comments to the Author**

1. If the authors have adequately addressed your comments raised in a previous round of review and you feel that this manuscript is now acceptable for publication, you may indicate that here to bypass the “Comments to the Author” section, enter your conflict of interest statement in the “Confidential to Editor” section, and submit your "Accept" recommendation.

Reviewer #1: All comments have been addressed

2. Is the manuscript technically sound, and do the data support the conclusions?

Reviewer #1: Yes

3. Has the statistical analysis been performed appropriately and rigorously? 

Reviewer #1: I Don't Know

4. Have the authors made all data underlying the findings in their manuscript fully available?

Reviewer #1: No

5. Is the manuscript presented in an intelligible fashion and written in standard English?

Reviewer #1: Yes

6. Review Comments to the Author

Reviewer #1: I think the authors adequately addressed all my comments. The limitations of the project have been outlined in the Discussion, and additional information on focus groups is also provided

7. PLOS authors have the option to publish the peer review history of their article (what does this mean?). If published, this will include your full peer review and any attached files.

Reviewer #1: No

---

## [Editor Report · Acceptance letter]

22 Mar 2023

PONE-D-22-29056R1 

Consumer preferences for telehealth in Australia: a discrete choice experiment 

Dear Dr. Savira:

I'm pleased to inform you that your manuscript has been deemed suitable for publication in PLOS ONE. Congratulations! Your manuscript is now with our production department. 

Kind regards, 

on behalf of

Dr. Francesca Ferrè 

Academic Editor

PLOS ONE